# Smart Home Control System Using VLC and Bluetooth Enabled AC Light Bulb for 3D Indoor Localization with Centimeter-Level Precision

**DOI:** 10.3390/s22218181

**Published:** 2022-10-26

**Authors:** Bo Xu, Babar Hussain, Yiru Wang, Hoi Chuen Cheng, Chik Patrick Yue

**Affiliations:** 1HKUST Shenzhen-Hong Kong Collaborative Innovation Research Institute, Shenzhen 518048, China; 2Department of Electronic and Computer Engineering, The Hong Kong University of Science and Technology, Hong Kong SAR 999077, China

**Keywords:** smart home, visible light communication (VLC), Internet of Things (IoT), visible light positioning (VLP), Bluetooth

## Abstract

Smart home systems (SHSs) are a modern lifestyle trend, changing daily lives in the most intuitive way. To connect and operate various smart devices under one system, an accurate, convenient, and secure control method is of utmost significance. Nowadays, most smart home control systems are based on radio-frequency (RF) technologies such as Bluetooth, Wi-Fi, and Zigbee. They tend to suffer from poor location accuracy in high-density environments due to the interference and noise in RF signals as well as their penetration through walls, leading to security vulnerabilities and low-precision control. To address these issues, this paper presents a smart home control system based on visible light communication (VLC), with enhanced security and accurate localization for precise and convenient control. The system includes an AC lightbulb, a smartphone running the position and control applications, and a cloud server with location-based access and a database of smart home devices. The design of the AC lightbulb integrates VLC and Bluetooth connectivity in a standard form factor for easy installation and plug-n-play capability. A smartphone camera-based 3D indoor positioning and orientation algorithm that allows precise control by pointing the smartphone the device is also presented. We demonstrate the feasibility of this system through prototype implementation and experimental verification.

## 1. Introduction

With the fast development of intelligent electronics, smart devices have dramatically increased in both variety and quantity, and they have been applied in locations such as homes, shopping malls, hospitals and factories. This tremendous growth has spawned the network technology called the Internet of Things (IoT) [1], which serves to exchange information among smart devices and support them in working as a system. IoT technology has dramatically contributed to the development of smart living concepts, such as smart industry, smart transportation, smart cities, and especially, smart home systems (SHSs) [2]. In SHSs, more and more domestic appliances are being designed and equipped with intelligent functions, such as timing, remote management, voice control, and adaptive adjustment with sensors [3,4], to improve ease of living. With such comprehensive intelligent features integrated into one system, users can adjust their living environments for the most comfort and safety.

However, the explosive growth in SHSs has brought several challenges, two of which are the efficient management of multiple smart devices and the guarantee of information security. To meet the first challenge, commercial SHSs adopt a hub as the center of the system and utilize networking technologies such as Wi-Fi, Bluetooth, ZigBee, and near-field communication (NFC) for communication among the smart devices and the controlling device [5,6]. Although these radio-frequency (RF)-based control methods simplify the control process, they suffer from poor control accuracy in the areas with a high number of devices due to the low positioning accuracy of meter level [7]. For example, users may find it difficult to distinguish the same type of device installed in neighboring rooms through a smartphone. This low indoor positioning accuracy is caused by electro-magnetic interference (EMI), signal reflection, environmental absorption, and noise in RF signals. The second challenge is information disclosure, which is caused by RF signals being able to easily penetrate through walls. Such leaks in information could threaten the safety of an SHS [8]. Although countermeasures to security issues such as anonymization, attribute access control systems, and cryptographic techniques [9], have been proposed, they are complex to implement. In addition, these solutions based on algorithms increase the development cost while decreasing the user convenience.

Visible light communication (VLC)-enabled smart home control system is a potential solution to the mentioned problems. VLC technology utilizes visible light as a medium of communication to transmit programmed data by modulating the intensity of the light source [10]. Due to the unique characteristics of visible light in terms of line-of-sight (LOS) capability, compatibility with existing lighting infrastructure, and absence of EMI, VLC is considered as a potential indoor communication method with high security and low cost.

In addition, visible light can be applied to localization with centimeter-level accuracy [11], which means indoor visible light positioning (VLP) could potentially solve the problem of low-accuracy indoor positioning of users in SHSs and provide location-based applications in control systems. The traditional indoor positioning algorithms utilize RF-based communication methods such as Wi-Fi [12], Bluetooth [13], and RFID [14] to calculate the user’s position. In [12], the received signal strength indicator (RSSI) of Wi-Fi signals between a transmitter and receiver pair is utilized to calculate the total number of people walking in one area. The data in Bluetooth low-energy beacon networks are collected and processed in a machine learning model to detect the user’s position with zone-level accuracy [13]. Reference [14] uses RFID readers to localize the position of the RFID tag, which is fused with laser data to localize the mobile robots. Compared with VLP technology, these algorithms utilizing RF signals suffer from the challenge of localization accuracy. They can only be used to account for the number of people or localize with zone-level accuracy, which is not competitive with the centimeter accuracy provided by VLP. As for the RFID, although it has accuracy competitive with that of VLP, it requires extra devices such as RFID tags and readers, which is not convenient for users. In [15], the data from vision sensors are fused into a Bayesian network to calculate and predict the occupancy of living buildings, which is also not able to provide the user with an accurate position and requires extra devices. In brief, VLP technology is superior in high-accuracy indoor positioning, and it is convenient for the user to acquire the information without adding extra devices.

However, VLC has implementation challenges that have prevented its widespread usage in real-world applications. VLC hardware currently reported in the literature is mainly based on experimental prototypes [16] and does not meet the requirements for practical deployment. To apply a VLC system in the real world, a compact size, low power consumption, compatibility with existing lighting infrastructure in homes, remote connectivity for management, and support of a smartphone as the receiver should all be taken into consideration. Therefore, there exists a need for developing a compact VLC lightbulb with a standard form factor, low manufacture cost, and plug-n-play functionality for user convenience. Moreover, supporting remote connectivity methods such as Wi-Fi and Bluetooth is also necessary for the VLC lightbulb design since they can dramatically improve the convenience of managing the lightbulbs. To compensate for the lack of acquiring high-accuracy indoor user positions in RF-based systems, implementing the VLP function through VLC lightbulbs and a smartphone is also important for real-world application.

With the mentioned limitations existing in the traditional RF-based control systems and VLC system, a motivational scenario of a VLC-enabled smart home control system can be presented. From the aspect of solving the limitations, the user can be facilitated if he or she can easily manage the working state of the target device without interference from the same devices installed in the neighboring rooms. In addition, a VLC lightbulb with a standard form factor that can provide plug-n-play functionality with remote control can bring convenience to the users. Simply using the smartphone as a VLC receiver to acquire the transmitted information can contribute to applying the VLC system in the real world. Moreover, the user will be interested in finding out his or her precise 3D position in SHSs with some location-based applications such as controlling a target device by pointing a smartphone in the corresponding direction or navigating a domestic robot to a specific position. In summary, the motivations for proposing this VLC-enabled smart home control system are as follows:(1)We aim to solve the low control accuracy problem and security issue in traditional RF-based smart home control systems.(2)We aim to solve the practical implementation problem of the VLC system and meet the requirements for real-world application.(3)We aim to solve the problem of low-accuracy indoor localization of users in SHSs by utilizing the VLP algorithm.

In this paper, we address the issues of accurate and secure management of smart home devices by presenting a practical VLC-based smart home control system using a VLC-integrated LED lightbulb previously proposed by us in [17], along with a 3D indoor localization algorithm based on VLP. In the application scenario presented in Figure 1, the VLC lightbulb transmits light signals containing a unique ID, which can be decoded by the smartphone. Then, the user can control the smart devices through Bluetooth with assistance from the cloud server and without signal interference from neighboring rooms. In this application scenario, the smart devices installed in the whole control region are divided into different rooms according to the position of the VLC lightbulb. As shown in Figure 1, there are three rooms forming the complete scenario, and the user in each room can only detect and control the smart devices existing in the same room. The control region division method not only solves the low control accuracy problem but also solves the challenge of security issues as the devices cannot be connected or controlled when the user is not in the specific room. The VLP algorithm utilizes the directional angles of the smartphone and the position of the lightbulb to calculate the user’s position and orientation. Our contributions are summarized as follows:(1)We propose a VLC-enabled smart home control system with the combination of Bluetooth communication. The proposed system aims to solve the limited control accuracy and security issues existing in traditional RF-based control systems. This system consists of a VLC lightbulb as a transmitter, a smartphone application-based receiver, and a back-end cloud server. These three key elements cooperate to provide precise and secure control of smart devices by dividing the whole control region into different rooms according to the position of the VLC lightbulb.(2)We present the design of a driverless AC-powered VLC lightbulb for the practical implementation of the VLC system. This VLC lightbulb is designed to solve the problems and meet the real-world application requirements for our proposed smart home control system. It is characterized by compact size, standard form factor, and plug-n-play functionality. Moreover, it supports wireless control via Bluetooth for remotely managing its working states and transmitted information, which brings convenience to users.(3)We propose a directional-angle-assisted 3D indoor VLP algorithm to acquire the precise 3D world coordinates of the user to compensate for the low user position accuracy in SHSs. With this proposed VLP algorithm, the user’s position can be calculated with centimeter-level accuracy. In addition, this algorithm can be adapted to orientation-based and location-based applications for extending diversity.

The remainder of this paper is organized as follows: In Section 2, we summarize related works on smart home control systems. The methodology and architecture of our proposed VLC-based smart home control system with 3D indoor positioning algorithm are presented in Section 3. In Section 4, we demonstrate the prototype and analyze the experimental results. Finally, we conclude the paper in Section 5.

## 2. Related Works

A smart home control system is an important subsystem of an SHS as it organizes intelligent appliances into a network and manages their working states. It aims to provide users with a convenient, power-efficient, and secure method to adjust their living environments. Considering this significant role, researchers are focusing on smart home control systems, with research emphases ranging from system architecture to significance, various control methods, and system applications. To provide users with better services, seven design principles for smart home control systems have been established for programming the software of the control system [18]. A power line communication (PLC)-assisted smart home control system [19,20] utilizes the power line to transmit control messages among smart devices and supply power considering the environment parameters collected from the wireless sensor network (WSN) to save energy. This system offers high accuracy control but suffers from low scalability as it relies on a fixed power line structure. An IoT-based smart home control system [21,22] includes WSN to provide environmental information, as well as a control system based on a gateway, cloud server, and smartphone app. This scalable control system offers a remote management function but suffers from poor control accuracy with unsolved security problems. Another type of smart home control system is supported by information fusion [23] and consists of an information fusion-based controller, power line, Bluetooth wireless networking, and computational units. It collects external factors and utilizes a fuzzy neural network (FNN)-based information fusion algorithm to determine the states of smart devices automatically. This system combines wireless and wireline networking for scalability and accuracy, although it lacks consideration for security. A smart home control system based on ZigBee [24,25] utilizes ZigBee with the IEEE 802.15.4 standard for communication and provides automatic energy management for energy saving. However, this system also faces accuracy and security problems. The security problem of SHS is widely discussed, along with potential solutions.

To date, only a few works have proposed the use of VLC in the context of smart homes. The idea of using purely VLC-based bidirectional communication for smart home connectivity was proposed in [26]. In [27], the authors presented an encryption mechanism to prevent the intrusion of adjacent adversaries in smart home settings. However, they did not discuss the related practical deployment scenarios and associated challenges. The security aspect of VLC-based smart home system implementation was discussed in [28], where an automated user authorization scheme based on proximity to visible light was presented for smart home settings. In addition to the lack of positioning accuracy, the presented design was based on a small prototype with a photodiode used as a receiver, making it impractical for integration with a smartphone and real-world deployment.

In the commercial application area, wireless-communication-supported smart control systems are also widely considered. Reference [29] introduces the plug load management system for energy saving, which comprises intelligent socket devices, a gateway, and a cloud server. The wireless communication among devices and the cloud server contributes to the users by measuring and analyzing device energy use. Reference [30] has mentioned a smart energy management system based on an IoT network. It uses the interconnected network to organize the modules and subsystems and implement the plug load automation. A smart energy management system controlling the sockets through ZigBee is proposed in [31]. It aims to reduce power consumption with a motion sensor and set the time of power usage. Compared with these smart energy control systems supported by RF-based communication methods, our proposed control system can provide more accurate control of devices with higher information security.

In summary, the typical methods that are adopted in smart home control systems include PLC, IoT network, information fusion, ZigBee, and VLC methods. Our proposed system utilizes a combination of VLC, BLE, and VLP technologies. The details of the works mentioned above are summarized in Table 1 with a comparison of our proposed system. The performances of these methods are compared with ours in Table 2. It can be noticed that the works based on RF networks lack high-accuracy control and system security, while wireline-based systems are poor at scalability. Our proposed system can provide users with an accurate and secure control system with high-precision 3D indoor localization.

## 3. Theoretical Analysis of Proposed Smart Home Control System

### 3.1. Architecture Design of Smart Home Control with Cloud Server

The VLC-based smart home control system consists of the hardware infrastructure made up of driverless AC-powered VLC lightbulbs with standard E27 sockets supporting direct plug-n-play capability. It also includes the supporting software which comprises a back-end cloud server storing the world coordinates of the lightbulbs and information of the smart devices in a look-up table (LUT) and a smartphone software application that implements the VLC decoding algorithm, Bluetooth-based control for the smart devices, and interfacing with the information stored in the cloud server. These three components network and manage various smart devices in the SHS.

The architecture and signal flows are presented in Figure 2. First, after the VLC lightbulb sends out the VLC signal containing a unique ID code, the user receives this signal through the smartphone app which captures the image of the VLC lightbulb as a rolling shutter pattern because of the CMOS image sensor (CIS)-based camera operation. Next, the decoding process is performed in the smartphone application. After the smartphone receives the rolling shutter patterns captured by the camara, the region of interest (ROI) which contains the patterns will be cropped and used in the threshold-determination process. The pixel values of the ROI-contained rolling shutter patterns will be read out, and the maximum and minimum values can be determined. Then the thresholds are calculated as the middle point of maximum and minimum values. The pixel values larger than the threshold are converted to “1” bits, and those smaller than the threshold are converted to “0” bits. Finally, the different binary streams are combined to form a complete data frame consisting of a preamble, payload (ID code), and error check sequence. The unique ID code contained in this data frame will be mapped to a uniform resource identifier (URI) database. Then the translation of ID code to URI is processed. The translated URI can be eventually used by the application software to interact with the cloud server on the Internet through wireless network communication methods such as Wi-Fi and LTE [32]. In the cloud server, the location data associated with each lightbulb are stored according to their ID code. Additionally, the Bluetooth MAC addresses of the smart devices installed in the same room as the lightbulb are recorded in terms of the ID code of the lightbulb. Therefore, each lightbulb has a pre-recorded file in the cloud server with its location data in a global map and the MAC address information of the smart devices co-located in the same room. After matching the ID code received from the smartphone, the cloud server can then provide the corresponding location data with the MAC addresses of the co-located smart devices. Since all smart devices are continuously broadcasting, the smartphone may detect many devices installed in different rooms in a home as the Bluetooth signal can penetrate through walls. With the returned MAC address, the smartphone can now identify which smart devices are installed in the same room as the VLC lightbulb and send out connection request only to these devices. Through the Bluetooth connection, the user can then manage and control these smart devices precisely.

### 3.2. Design of VLC LED Lightbulb

In our proposed system, the core hardware infrastructure is made up of the VLC LED lightbulbs which serve as beacons and generate the VLC signals. Considering that the typical ceiling height of housing is around 3 m and suitable illumination intensity needs to be more than 300 lx, the VLC link distance should cover this range and the total output power of the lightbulb must provide sufficient illumination for lighting. Furthermore, most VLC-enabled light sources require an external modulator, which is inconvenient for installation for standard ceiling light sockets.

To meet the above requirements, our proposed solution is shown in Figure 3. The proposed driverless VLC lightbulb consists of four parts: an AC–DC converter circuit, a DC–DC converter circuit, a Bluetooth Low Energy (BLE) control circuit [33], and 24 LEDs connected in series. In the AC–DC converter circuit, a resistor–capacitor (RC) parallel circuit is connected in series with a bridge rectifier, and the impedance of capacitor C1 can be employed as a voltage divider to decrease the large AC input voltage down to the desired output voltage for supplying loads. The resistor R1 connected in parallel with C1 is the bleeder resistor, which helps to neutralize the remaining charges in the capacitors during the power-off state, while they can be treated as open circuits during the power-on state, guaranteeing less power waste. The capacitance of C1 can be determined using the following equation:(1)$$X_{c}=\frac{1}{\left(2\pi \mathrm{fC}\right)}$$
where $X_{c}$ is the capacitive reactance of capacitor C1, C is the capacitance and f is the frequency of AC input voltage. The required value of $X_{c}$ for dividing the input voltage can be determined as follows:(2)$$I_{\mathrm{LED}}=\frac{V_{\mathrm{total}}}{X_{c}},$$
where $V_{\mathrm{total}}$ is the decrease value of input 220 V AC voltage using the voltage divider and is equal to 148 V, and $I_{\mathrm{LED}}$ is the driving current of the LED string and is directly provided by the LED datasheet as equal to 130 mA. Since the target electric power of this lightbulb is 10 W and the turn-on voltage of each LED is 3 V, the total number of LEDs required can be calculated by the following equation:(3)$$P_{\mathrm{LED}}=N_{\mathrm{LED}}\times V_{\mathrm{LED}}\times I_{i\_R},$$
where $P_{\mathrm{LED}}$ is the target electric power of this lightbulb and is equal to 10 W, and $V_{\mathrm{LED}}$ is the 3 V supply voltage of each LED. Therefore, the total number of required LEDs is 24. The bridge rectifier converts the decreased AC input voltage to a 72 V DC output, and the following RC filter connected in parallel with the rectifier smooths the ripple of this 72 V output voltage which is utilized to supply both the LED series and the subsequent control block. Therefore, the calculated value of C1 is around 2.7 uF for decreasing the 220 V AV input voltage to 72 V. The following RC filter connected in parallel with the rectifier smooths the ripple of this 72 V output voltage. This output is then supplied to both the LED string and the VLC control module.

The DC–DC converter circuit and BLE control circuit constitute a control block realizing the function of a normal external modulator, but with a more compact scale that can be integrated into the shell of a lightbulb. In the DC–DC converter circuit, the first low dropout regulator (LDO) decreases the 72 V DC input voltage to 5 V and supplies the pre-driver with resistors R3 and R4 serving as an output voltage controller. The second LDO converts 5 V to 3.3 V DC voltage and guarantees that the BLE module works at the proper voltage. Capacitors C3, C4, and C5 help to smooth input and output DC voltage. The BLE control circuit contains a pre-driver, BLE module, and high-threshold power MOSFET. The BLE module [34] loads the VLC ID code stored in the embedded flash memory to the RAM and sends out pulse-width modulation (PWM) waveforms through the serial peripheral interface (SPI). Furthermore, the VLC ID code can be changed wirelessly through Bluetooth using a programming application [34]. To meet the requirement of output power driving capability, a CMOS inverter-based driver is inserted between the I/O of the BLE module and the power MOSFET. The power MOSFET acts as a switch controlling the on/off states of the LEDs under a switching speed of 8–16 kHz, and it is chosen to sustain a drain-to-source voltage of up to 72 V during the LED off state.

Figure 4 shows the simulated results of the designed high-power VLC lightbulb, showing the AC input voltage; LED supply input voltage; modulation signal, which is the stored VLC signal in the BLE module; and LED driving current. The zoomed-in waveforms reveal that the simulated supply voltage for the LEDs is around 74 V and the average value of the driving current is 130 mA, and these values are consistent with the design targets. It can also be noticed that the waveform of the driving current changes in correspondence with the modulation signal.

### 3.3. Smart Home Control with Directional Angle Data Assisted 3D Indoor VLP Algorithm

Security is a major concern in several application scenarios, particularly in residential environments. The traditional RF-based communication technologies such as Wi-Fi, Bluetooth, and ZigBee face a security challenge since the signal can penetrate through walls, allowing someone outside the house premises to control the devices inside. If the smart devices are managed with position-based authorization, i.e., the user can only control the device by being at a specific area inside the house defined via precise 3D position coordinates, the security of the system can be significantly improved. In addition, high-precision positioning can further help in simplifying control by using the orientation angle of a smartphone for distinguishing multiple identical smart devices in close proximity. For instance, the user can directly point the phone at the target device to establish a connection based on the current location and orientation.

Inspired by the two propositions above, we propose a smart home control system based on an orientation-supported 3D positioning algorithm [35] to further increase the precision of indoor positioning. The proposed 3D positioning algorithm relies on the principle of projective geometry, where the smartphone camera takes a picture of the LED lightbulb in the ceiling and the corresponding projection of the light on the image is used to determine the relative position of the smartphone w.r.t the LED light [36], as shown in Figure 5. The positioning algorithm follows the principle of the pinhole camera model, which is given as follows:(4)$$s\left[\begin{array}{c} u\\ v\\ 1 \end{array}\right]=K\left[\begin{array}{c} X_{C}\\ Y_{C}\\ Z_{C} \end{array}\right]$$
where $X_{C}$, $Y_{C}$, and $Z_{C}$ represent 3D coordinates of the LED light w.r.t the camera coordinate frame. The camera transforms these 3D points into 2D px coordinates u and v, with s being the scaling factor, on the image plane using the transformation matrix K. The matrix K, called the camera intrinsic matrix, can be expanded to rewrite Equation (4) as follows:(5)$$s\left[\begin{array}{c} u\\ v\\ 1 \end{array}\right]=\left[\begin{array}{cccc} f_{x} & \gamma & u_{0} & 0\\ 0 & f_{y} & v_{0} & 0\\ 0 & 0 & 1 & 0 \end{array}\right]\left[\begin{array}{c} X_{C}\\ Y_{C}\\ Z_{C} \end{array}\right]$$
where $f_{x}$ and $f_{y}$ are the focal lengths of the x-axis and y-axis, respectively; ($u_{0},\;v_{0}$) are the center coordinates of the image plane; and γ is the skew coefficient between the x-axis and y-axis. Since the information on the LED light’s position is in the 3D world coordinates ($X_{W}$, $Y_{W}$, $Z_{W}$), a transformation needs to be performed to convert from the world frame to the camera frame. This transformation is performed by inserting a 3 × 3 rotation matrix $R_{3\times 3}$ and a 3 × 1 translation matrix $T_{3\times 1}$ in Equation (5) as follows:(6)$$z_{c}\left[\begin{array}{c} u\\ v\\ 1 \end{array}\right]=\begin{array}{c} \left[\begin{array}{cccc} f_{x} & \gamma & u_{0} & 0\\ 0 & f_{y} & v_{0} & 0\\ 0 & 0 & 1 & 0 \end{array}\right] \end{array}\begin{array}{cc} \overset{R_{3\times 3}}{\overbrace{[\begin{array}{ccc} r_{1} & r_{2} & r_{3}\\ r_{4} & r_{5} & r_{6}\\ r_{7} & r_{8} & r_{9} \end{array}}} & \overset{T_{3\times 1}}{\overbrace{\begin{array}{c} {\;x}_{0}\\ {\;y}_{0}\\ {\;z}_{0} \end{array}]}} \end{array}\begin{array}{c} \left[\begin{array}{c} X_{w}\\ Y_{w}\\ Z_{w}\\ 1 \end{array}\right] \end{array}.$$

It is important to note that the translation matrix $T_{3\times 1}$ given in Equation (6) represents the 3D coordinates of the camera w.r.t the LED light assuming the coordinates of the LED to be (0, 0, $Z_{w}$), where $Z_{w}$ is the height of the LED light measured from the ground up. Therefore $T_{3\times 1}$ becomes the 3D position of the smartphone and is treated as the target unknown in Equation (6). On other hand, $R_{3\times 3}$ can be calculated using the built-in inertial sensors of the smartphone, which provide the three smartphone orientation angles, i.e., roll ($\varphi_{x}$), pitch ($\varphi_{y}$), and azimuth ($\varphi_{z}$), as shown in Figure 5 on the left. The expansion of $R_{3\times 3}$ based on these smartphone orientation angles is given by the following:(7)$$R_{3\times 3}=\left[\begin{array}{ccc} {\cos\varphi }_{z} & {\sin\varphi }_{z} & 0\\ {\sin\varphi }_{z} & {\cos\varphi }_{z} & 0\\ 0 & 0 & 1 \end{array}\right]\;\times \;\left[\begin{array}{ccc} 1 & 0 & 0\\ 0 & {\cos\varphi }_{x} & {\sin\varphi }_{x}\\ 0 & -{\sin\varphi }_{x} & {\cos\varphi }_{x} \end{array}\right]\;\times \;\left[\begin{array}{ccc} {\cos\varphi }_{y} & 0 & {\sin\varphi }_{y}\\ 0 & 1 & 0\\ -{\sin\varphi }_{y} & 0 & {\cos\varphi }_{y} \end{array}\right]$$

To calculate $T_{3\times 1}$ using the relation in (6), the 2D pixel coordinates of the light’s projection on the image, i.e., (u, v), and the 3D world coordinates of the LED light, i.e., $X_{w}$, $Y_{w}$, and $Z_{w}$, along with the calculated rotation matrix $R_{3\times 3}$ are put in the equation. The scaling factor s can be calculated using the relation in Equation (8), where a is the length of the semi-major axis of the ellipse at the image plane and r is the diameter of the LED lightbulb.
(8)$$s=\frac{a}{r}$$

## 4. Smart Home Control System Prototype and Performance Evaluation

### 4.1. Prototype of VLC Lightbulb

The detailed layout design of the PCB for the proposed AC-powered VLC lightbulb is presented in Figure 6. Figure 6a presents the layout design of the power management board and LED string, which consists of the AC–DC converter circuit and 24 LEDs connected in series. The diameter of this PCB board is 9 cm for fitting the size of a standard lightbulb enclosure. The layout design of the BLE control board, which comprises the DC–DC converter circuit and BLE control circuit, is shown in Figure 6b. The size of this PCB board is 3 × 5 cm, and this compact design targets the removal of the traditional large external modulator for practical applications. Moreover, heat sinks are applied to the bottom of LDO1 and the power MOSFET for heat emission. As for the manufacturing cost, this VLC lightbulb prototype is cost-effective, and the cost can be reduced in large-scale manufacturing since the prices of adopted components and PCB boards decrease dramatically with large-scale order numbers. This relatively low cost of the VLC lightbulb also guarantees its possibility for real-world applications.

The proposed high-power VLC lightbulb is fabricated and retrofitted in a commercial 9 cm diameter AC lightbulb enclosure with a standard E27 socket for practical implementation, as shown in Figure 7. The AC–DC converter circuit and LED string are combined on one PCB board while the BLE control board is affixed on its back side. Both PCBs are enclosed in the lightbulb shell.

The measured results of this designed VLC lightbulb are summarized in Table 3. To support the operation of 24 LEDs connected in series, the designed total operating voltage for the LEDs is 72 V. As presented in Figure 4, the simulated operating voltage of this VLC lightbulb presented is 74 V. The measured actual operating voltage of LEDs is 74.2 V, which is close to the design target and consistent with the measured result. This voltage is high enough to maintain the LEDs working in the suitable region. Another important factor for estimating the working states of this VLC lightbulb is the driving current of LEDs. According to the datasheet, the driving current of a single LED under the operating voltage of 3 V is 130 mA. In the simulation, the average driving current of this LED series is 130 mA. The measured current through the LED string is 139 mA, which is also consistent with the simulation result. As for the total electric power that this VLC consumed, our design target is 10 W for guaranteeing the indoor illumination requirements as the illumination is proportional to the electric power. The measured electric power is 10.3 W and meets the design target. With this electric power, the measured illuminance at a distance of 1.5 m is 362.5 lx, which is higher than the illumination requirement of 300 lx in the domestic environment.

The typical ceiling height in residential buildings is around 3 m and requires more than 300 lx for illumination. To meet these real-life application requirements, we studied the relationships between the diffuser diameter and VLC link distance as well as illuminance and distance. The setup for measuring the VLC link distance is shown in Figure 8, and the corresponding measured results are presented in Figure 9. It can be noticed that the VLC link distance is proportional to the diffuser diameter. Thus, a 20 cm lamp housing is applied outside of the lightbulb to reach up to a 3.4 m VLC link distance. The measured illuminance at 1.5 m from the lightbulb is 362.5 lx, which also meets the requirement for residential lighting.

### 4.2. Performance Evaluation of VLC-Based Smart Home Control System

The demonstration of a VLC-based smart home control system consisting of three VLC lightbulbs and the user smartphone app is presented in Figure 10. These three VLC lightbulbs are programmed with different ID codes, named Light ID 1, 2, and 3 separately, which are stored in the memory unit of the BLE module. Then, we record the corresponding location data and Bluetooth MAC addresses of the smart devices in the cloud server with the ID code of the lightbulb. The cloud-server-based architecture provides several advantages. Firstly, it guarantees the safety of data storage; secondly, it simplifies the management of the SHS with remote control function; lastly, the modification of the smart home network can be conveniently processed on the cloud server by changing the stored information in the LUT.

It can be seen in Figure 10 that the location of the lightbulb with ID code 1 is the kitchen and there are smart devices in the kitchen that can be controlled. ID code 2 identifies the dining room, where three controllable devices are located. Lastly, ID code 3 indicates the bedroom where three devices can be managed. When the user moves from the kitchen where lightbulb 1 is installed to the dining room where lightbulb 2 exists, the location and device information provided on the smartphone screen changes from kitchen to dining room correspondingly. At the same time, the Bluetooth connection between the smartphone and smart devices is updated automatically when a different VLC ID code is received. Therefore, the whole control region is divided into separate small regions according to the position of the VLC lightbulb. This region division method can solve the low control accuracy problem by avoiding interference from devices in other rooms. Moreover, it guarantees the safety of the proposed system as the information of devices in one region will not be provided to users outside of this region. In other words, the user can only manage the smart devices in a specific room after entering the specific room.

When considering the evaluation of the proposed system in real-world applications, possible concerns about the operation of this application and light interference are raised. This smartphone application is not left running in the background all the time for energy effectiveness. It is only utilized to choose the working states of smart devices instead of maintaining the operation of these devices, and the user can exit this smartphone app after the working states of smart devices are decided. As for the interference from other illuminating infrastructures, it will not severely affect the performance of our system. The background light is several orders of magnitude less than the LED light signal itself. When the camera sensitivity is set to be so low that only a direct LED light signal can be detected by the image sensor by changing the ISO of the camera, the background light is automatically filtered out to eliminate the interference. As for the light from other light sources nearby that might have the same strength as the VLC lightbulb, it is detected in the same way as the VLC lightbulb. However, during the signal decoding stage, such lightbulbs are filtered out by the software-supported receiver as regions outside of the ROI since they do not provide the rolling shutter patterns for carrying the signals.

### 4.3. Verification of Directional Angle Data Assisted 3D Indoor VLP Algorithm

We verified the proposed smart home control system based on the direction-angle-supported 3D visible light positioning algorithm through an experiment. The experiment setup was the same as that shown in Figure 8, and the horizontal area was 1.5 × 1.5 m. The VLC lightbulb was mounted at a height of 1.9 m with a diffuser diameter of 20 cm. The camera resolution of the smartphone used in this experiment was 1920 × 1080 and the exposure time was 1/3000 s with ISO 100. The smartphone was placed in the delineated area, which was divided into a 20 cm × 50 cm grid. The origin position in this experimental area was the projected point of the lightbulb at the horizontal plane. For the direction angle, the pitch angle $\varphi_{x}$ and roll angle $\varphi_{y}$ were limited from −40° to 40°, and the azimuth angle $\varphi_{z}$ varied from −60° to 60°.

The smartphone was placed at four different positions with different direction angles, and the captured images at these four positions with rolling shutter patterns are shown in Figure 11. It is obvious that when the phone rotates and moves, the captured image changes from a circle to an ellipse and varies in area, which is quantized and applied to Equation (6) to calculate the 3D world coordinates of the camera. The measured directional angles and coordinates of the camera are presented in Table 3. To evaluate the proposed direction angle data supported 3D positioning algorithm, we calculate the positioning error rate (PER) using the function adopted in [37]:(9)$$\mathrm{PER}\left(\%\right)=\frac{\sqrt{\Delta x_{0}{}^{2}+\Delta y_{0}{}^{2}+\Delta z_{0}{}^{2}}}{\sqrt{x_{0}{}^{2}{+y}_{0}{}^{2}{+z}_{0}{}^{2}}}\;\times \;100\%,$$
where ($x_{0}$, $y_{0}$, $z_{0}$) are the measured world coordinates of the camera and ($\Delta x_{0}$, $\Delta y_{0}$, $\Delta z_{0}$) are the calculated absolute values of the positioning error. From Table 4, it is obvious that the 3D world coordinates of the user can be obtained using the VLC lightbulb with a PER of approximately 7%. In terms of precision, this PER translates into a positioning error of less than 15 cm, which is sufficient for accurate and precise control of most smart home devices in close proximity.

## 5. Conclusions

In this work, we addressed two of the main concerns of modern smart home systems, namely the efficient and convenient management of a high density of smart home devices and the improvement of the security of the system. We presented a smart home system architecture that utilizes VLC-enabled LED lighting as location beacons and a smartphone with a camera as the receiver. The system also includes a control application running on the smartphone and a cloud server with a location and connection database of the smart home devices. The proposed system can accurately identify the user’s location and orientation to control a smart home device precisely and securely. To address the practical deployment considerations of VLC, we presented the design of a 10 W AC-powered LED lightbulb that integrates VLC functionality with Bluetooth-based wireless control in a standard form factor with an E27 socket. The lightbulb achieves a VLC link distance of up to 3.4 m while maintaining illumination of over 300 lx at a distance of 1.5 m. In addition, we presented an algorithm for calculating the 3D position and orientation of the smartphone using the presented lightbulb and verified the positioning accuracy through measurements, where the system was shown to achieve an average PER of less than 7%.

However, there are some limitations existing in our proposed VLC-enabled smart home control system which can be addressed in its future iterations. The first limitation is that this VLC lightbulb can be blocked by obstructions when the user is moving in the room. This blocking of visible light may result in incomplete captured patterns or even no patterns for the smartphone camera and lead to difficulty in the decoding process. To solve this limitation, we plan to adopt an image processing algorithm that can decode the transmitted signals with incomplete captured patterns in our future work. In addition, the use of multi-sensor fusion to solve the total blocking issue and further improve system performance is also a possible direction for the future iteration of this system. The second limitation is that the proposed smart home control system only combines Bluetooth-supported devices and lacks other types of communication methods such as Wi-Fi. At present, a large number of smart devices in the market support Wi-Fi-based communication networks. Therefore, combining Wi-Fi-supported devices into our proposed control system is a significant direction for increasing the possibility of real-world application. Moreover, we plan to expand the scope of the proposed system to include robots as autonomous SHS monitoring agents, utilizing the highly accurate locations from the LED lights to help the robots with navigation and control planning.

## Figures and Tables

**Figure 1 sensors-22-08181-f001:**
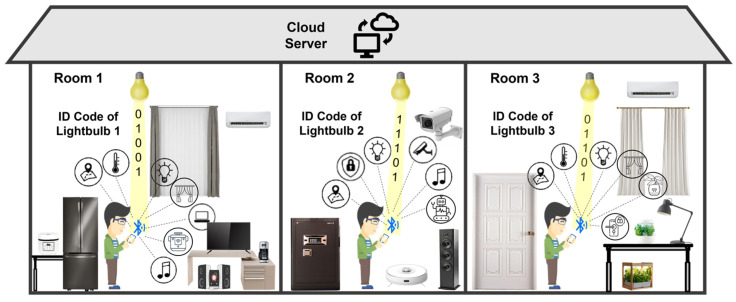
Application scenario of proposed VLC-based smart home control system.

**Figure 2 sensors-22-08181-f002:**
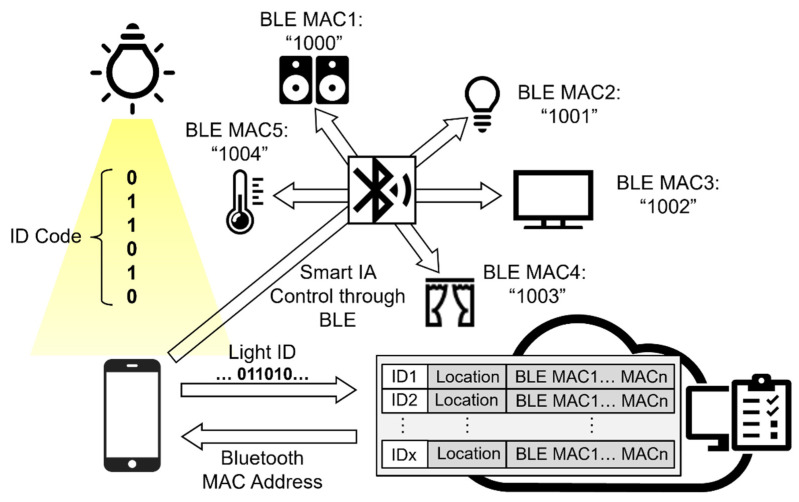
Architecture of proposed smart home control system.

**Figure 3 sensors-22-08181-f003:**
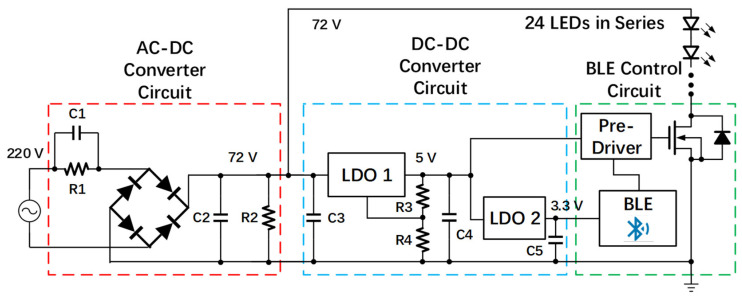
Schematic of high-power LED lightbulb with VLC functionality.

**Figure 4 sensors-22-08181-f004:**
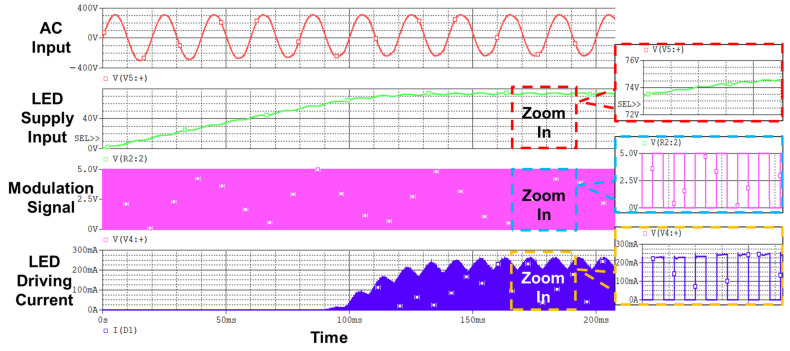
Simulated waveforms of designed VLC-based LED lightbulb.

**Figure 5 sensors-22-08181-f005:**
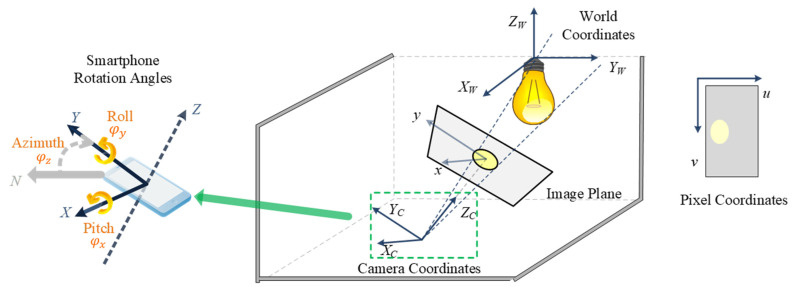
Imaging geometry of proposed direction angle data supported 3D positioning system [34].

**Figure 6 sensors-22-08181-f006:**
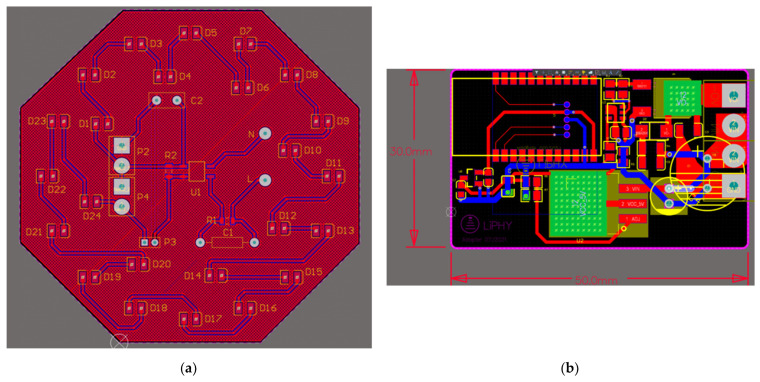
Details of PCB for the designed VLC LED lightbulb: (**a**) layout of LED and power management board; (**b**) layout of BLE control board.

**Figure 7 sensors-22-08181-f007:**
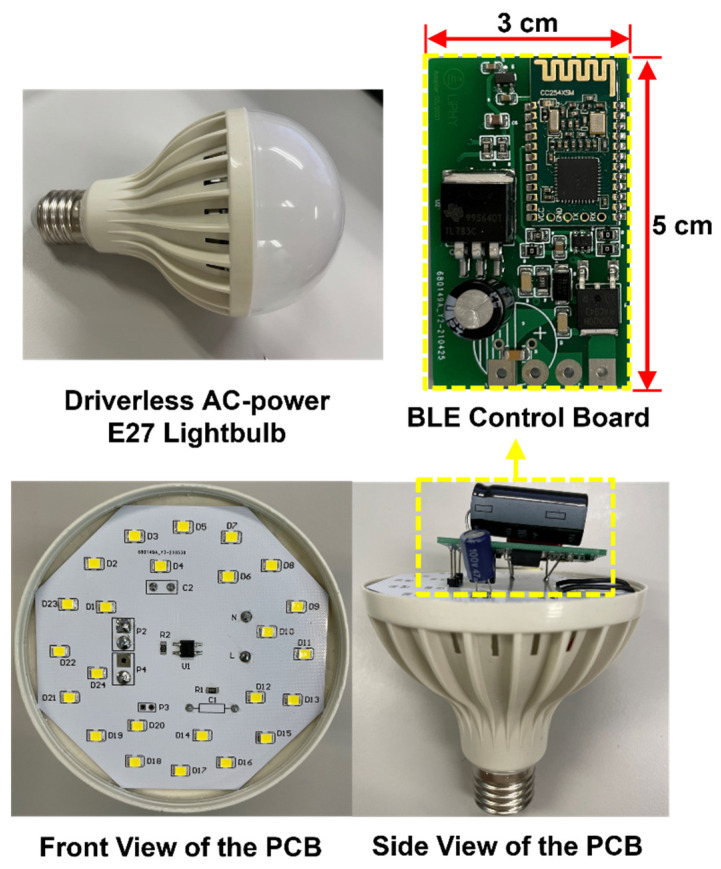
Prototype of VLC LED lightbulb.

**Figure 8 sensors-22-08181-f008:**
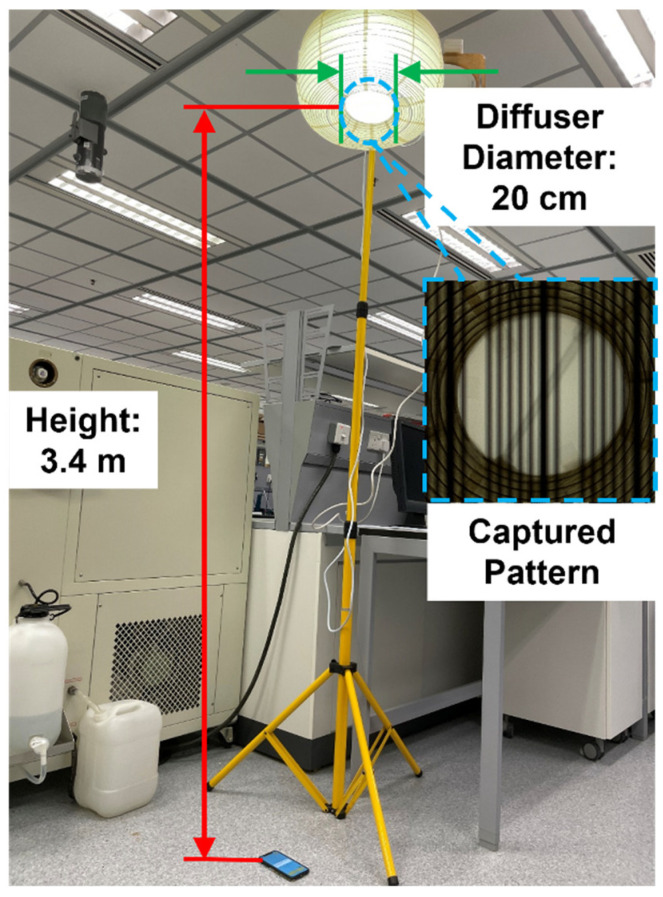
Experiment setup for measuring the VLC link distance.

**Figure 9 sensors-22-08181-f009:**
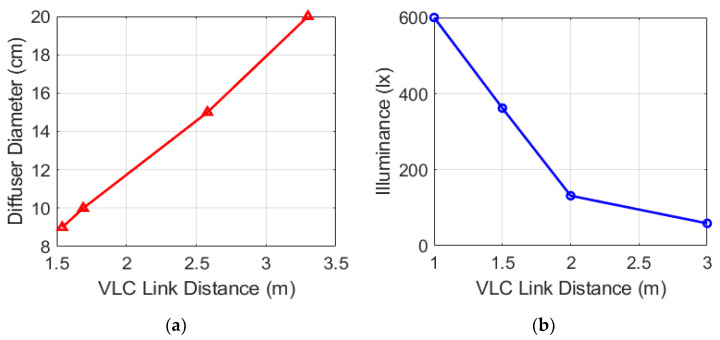
Impact of diffuser diameter and illuminance of lightbulbs on the VLC link distance: (**a**) diffuser diameter versus VLC link distance; (**b**) illuminance versus VLC link distance.

**Figure 10 sensors-22-08181-f010:**
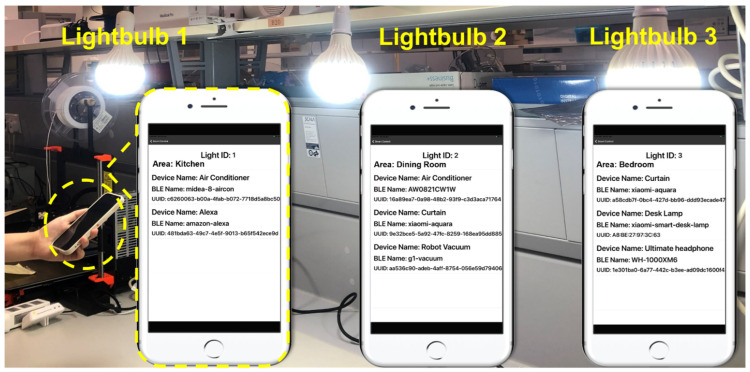
Demonstration of proposed VLC-based smart home control system [17].

**Figure 11 sensors-22-08181-f011:**
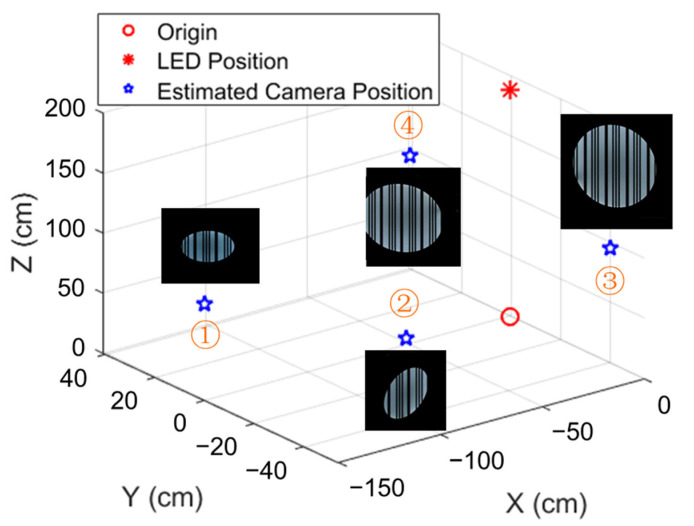
Measured 3D world coordinates of camera with images captured at four positions.

**Table 1 sensors-22-08181-t001:** Summary of related works on smart control systems.

Reference	Year of Publication	Adopted Technology	Communication Method	Core Devices
[19,20]	2015, 2014	PLC + WSN	PLC + ZigBee	ZigBee coordinator, power line, PLC TRx
[21,22]	2017	IoT + big data	Wi-Fi	Microcontroller, sensors, cloud server, smartphone
[23]	2008	Information fusion	Bluetooth + PLC	IF-based controller, power line, computational unit
[24,25]	2010	ZigBee	ZigBee	RF-based devices, sensors
[26,27,28]	2019	VLC	VLC	Evaluation board, smart home gateway, smartphone
This Work	/	VLC + VLP	VLC + BLE + Wi-Fi/LTE	VLC lightbulb, cloud server, smartphone

**Table 2 sensors-22-08181-t002:** Comparison of typical methods adopted in smart home control systems.

Typical Methods		PLC	IoT	Information Fusion	ZigBee	VLC	VLC + BLE + VLP
System Scalability		Low	High	Middle	High	Low	High
User Convenience		Middle	High	High	Middle	Low	High
Control Accuracy		High	Low	Middle	Low	Middle	High
Considered	Security Issue	No	No	No	No	Yes	Yes
	Indoor Positioning	No	No	No	No	No	Yes

**Table 3 sensors-22-08181-t003:** Measured working state of proposed VLC lightbulb.

Operating Voltage of LEDs	Driving Current of LEDs	Total Electric Power	VLC Link Distance	Illuminance at 1.5 m
74.2 V	139 mA	10.3 W	3.4 m	362.5 lx

**Table 4 sensors-22-08181-t004:** Measured direction angle, coordinates, and positioning error of camera.

Location	Direction Angles	$x_{0}$ (cm)	$y_{0}$ (cm)	$z_{0}$ (cm)	PER
Position 1	$\varphi_{x}$ = 40°, $\varphi_{y}$ = 0°, $\varphi_{z}$ = 0°	−151	0	81	6.72%
Position 2	$\varphi_{x}$ = −23°, $\varphi_{y}$ = −39°, $\varphi_{z}$ = 90°	−117	−54	88	6.87%
Position 3	$\varphi_{x}$ = −18°, $\varphi_{y}$ = −4°, $\varphi_{z}$ = 90°	−10	−49	109	6.23%
Position 4	$\varphi_{x}$ = 20°, $\varphi_{y}$ = 8°, $\varphi_{z}$ = 90°	0	41	84	7.41%

## Data Availability

Not applicable.

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
