# Peer review of "Smart Home Control System Using VLC and Bluetooth Enabled AC Light Bulb for 3D Indoor Localization with Centimeter-Level Precision"

_sensors, 2022, doi:10.3390/s22218181_

Round 1

Reviewer 1 Report

This paper introduces a control system for smart homes by using visible light communication. This control system is supposed to enhance security and guarantee accurate localization for precision control. This paper considers two main challenges to the growth of SHSs. The first one is related to information security when using multiple smart devices. The authors mentioned that the commercial solutions consider sung a single hub as a center for network technologies. The second challenge is related to information disclosure because the RF signals can penetrate walls, and this will threaten SHSs safety. The VLC can be used to transmit the programmed data by modulating the intensity of the light source. But the VLC hardware is mainly used for experimental prototypes, and it cannot be used for commercial deployments.

These challenges are behind the motivation of the authors to introduce their control system using VLC. After reading this paper, I have the following comments:

Major Comments:

-       The clear motivation behind introducing this paper is an information security and data privacy. This is clear from the first four paragraphs in the Introduction section. It is good to see two to three items in a list as a summary of the motivation behind introducing this paper.

-       A motivational scenario is urgently needed in the introduction section.

-       Line 82 to 84: The authors mentioned that “In this paper… by presenting a practical VLC-based smart home control system using a VLC-integrated LED light bulb [11],…”. Once you mentioned a reference here, so it is not your own contribution. You have to mention that you only used/utilized the VLC-integrated LED light bulb [11]. It is not proposed by you.

-       The main contributions introduced from line 91 to line 108 are so confusing and long sentences. The contributions mentioned here cover many challenges that are not listed in the motivational part. You need to be more specific. You have to focus on only two to three challenges and present the solution to them.

-       In Figure 1, the internal communication technology between the user and all the appliances is Bluetooth. You are still using RF technology in your application scenario. The Bluetooth range is about 10m. Thus, you haven’t solved the challenges.

-       The details about how the data will be transmitted to the cloud server are missing.

-       The details about the unique ID used by the lightbulb to transmit the light signals to the user. How this ID will be created? By default, each device in the IoT network has a unique identity. Thus, what is your contribution?

-       The lightbulb in your application scenario is a single point of bottleneck and attack.

-       The design of the VLC LED lightbulb mentioned in section 3.2 is redundant here. Moreover, it is not your own contribution to mention it under your proposed system.

-       The proposed sentence is not proven secure.

-       I see that the mentioned contributions in the introduction section have not been achieved in the paper.

Minor Comments:

There are many typos and grammatical issues in this paper. Please, revise the whole paper.

Reviewer 2 Report

In this paper, authors propose a smart home control system via the combined usage of VLC and RF 91 communication. They start conducting an extensive literature review, to then focus on the adopted methodology and a simulation study in which they compared performances of system implemented. Author has used directional angle-assisted 3D indoor VLP algorithm to calculate the world coordinates of the user.   In Section 4 they analyzed the prototype and experimental results. Finally, they focus on conclusions.

Although the addressed topic is current and interesting, the paper cannot be considered for publication. I recommend a Major revision. There are indeed a few major issues that need to be addressed:

1. The introduction does not help the reader understanding the main topic. There is a lot of generic (sometimes even too generic) information and it does not provide enough insights on the actual problem of interest. Some more references are also required to facilitate the comprehension. I suggest authors try to improve the structure of this section.

2. The literature review section essentially lists a number of works, one by one. It is really difficult to follow what authors want to communicate. There is the need of a table or an additional paragraph where authors summarize the contribution of each listed paper. For example, for each reference they could list the year of publication, the adopted methods, the data set they considered and their main conclusions. They could also group papers according to eventual similarities.

3. The simulation study needs some improvements. In particular, it would be better to introduce all the competing methods at the beginning (including the innovative proposed method) and then compare their performances instead of reporting a single table for each of them.

4.There is no actual discussion in the “Smart Home Control System Prototype and Performance Evaluation” they have to discussed in more detail about working state of proposed VLC lightbulb.

5. Language needs improvement and paragraphs need to be better defined. There are a few typos and several sentences are also quite confusing.

6. Sometimes citations are wrongly reported, please check them all.

Reviewer 3 Report

MDPI Sensors Journal (Manuscript ID: sensors-1942122)

Comments to the Author

This paper investigates the issues of secure, efficient and convenient management of smart home devices by presenting a practical VLC-based smart home control system using a VLC-integrated LED light bulb, along with a 3D indoor localisation algorithm based on VLP. It is an interesting topic and the paper studies the concept clearly. However, there are several points need to be addressed to improve the quality of the manuscript.

Suggestions to improve the quality of the paper are provided below:

1) Currently, the scope of the existing literature review is only focused on the residential home context. I strongly suggest that the authors expand the scope of the literature review to also include smart control systems in the commercial setting as many of the technologies used are identical with many common challenges. Please refer to the following works for commercial context.

Kandt, Alicen J., and Margarete R. Langner. Plug Load Management System Field Study. No. NREL/TP-7A40-72028. National Renewable Energy Lab.(NREL), Golden, CO (United States), 2019.

Tekler, Z. D., Low, R., Yuen, C., & Blessing, L. (2022). Plug-Mate: An IoT-based occupancy-driven plug load management system in smart buildings. Building and Environment, 109472.

Park, Sunghoi, et al. "Design and implementation of smart energy management system for reducing power consumption using ZigBee wireless communication module." Procedia Computer Science 19 (2013): 662-668.

2) Since the proposed system is tested in an experimental setting, please address the following points on how it can be generalised to real-world settings:

·       If the smartphone app is used to capture the image of the VLC lightbulb, does it mean that the app must be left running the background all the time? How does it impact the power consumption of the device?

·       What happens if the user’s smartphone is kept in their pockets? Would the detection of the VLC lightbulb fail in this case?

·       Have you considered the latency between the point at which the user takes out his/her phone to capture the VLC ID code and the point at which the user is able to control the devices in the room?

·       What is the estimated cost of the VLC LED lightbulb?

·       How does the brightness of the room or nearby light sources affect the detection accuracy of the system when trying to capture the VLC ID code?

3) Given that the proposed smart home control system is equipped with an indoor positioning algorithm, the authors should spend some time providing a brief review about the different indoor positioning algorithms that have been proposed in the literature. Many of these technologies uses wireless signals such as RFID, Wifi, Bluetooth Low Energy, and cameras to accurately locate an occupant in the building for different applications. Please refer to the following works as a starting point.

Depatla, Saandeep, Arjun Muralidharan, and Yasamin Mostofi. "Occupancy estimation using only WiFi power measurements." IEEE Journal on Selected Areas in Communications 33.7 (2015): 1381-1393.

Tekler, Z.D., Low, R., Gunay, B., Andersen, R.K. and Blessing, L., 2020. A scalable Bluetooth Low Energy approach to identify occupancy patterns and profiles in office spaces. Building and Environment171, p.106681.

Liu, Dixin, et al. "Measuring indoor occupancy in intelligent buildings using the fusion of vision sensors." Measurement Science and Technology 24.7 (2013): 074023.

Hahnel, Dirk, et al. "Mapping and localization with RFID technology." IEEE International Conference on Robotics and Automation, 2004. Proceedings. ICRA'04. 2004. Vol. 1. IEEE, 2004.

4) Please spend some time to think about the limitations of the existing system and discuss about how these limitations can be addressed in future iterations of this system.

5) Some minor feedback to also take note off:

·       For Figure 4, the zoom-ed in portions of the figure are overlapping with the previous graphs. I suggest that the authors do not do that to avoid over-cluttering the figure.

Round 2

Reviewer 1 Report

Good job.

Reviewer 2 Report

The author has incorporated all the comments.

Reviewer 3 Report

Thank you for addressing my concerns and comments carefully. The current version of the manuscript is ready for publication. Great work!